# Design and Implementation of Embedded-Based Vein Image Processing System with Enhanced Denoising Capabilities

**DOI:** 10.3390/s22218559

**Published:** 2022-11-07

**Authors:** Jongwon Lee, Incheol Jeong, Kapyol Kim, Jinsoo Cho

**Affiliations:** Department of Computer Engineering, Gachon University Global Campus, Seongnam 13120, Korea

**Keywords:** vein projection, vein detection, digital hair removal, embedded system, image processing

## Abstract

In general, it is very difficult to visually locate blood vessels for intravenous injection or surgery. In addition, if vein detection fails, physical and mental pain occurs to the patient and leads to financial loss in the hospital. In order to prevent this problem, NIR-based vein detection technology is developing. The proposed study combines vein detection and digital hair removal to eliminate body hair, a noise that hinders the accuracy of detection, improving the performance of the entire algorithm by about 10.38% over existing systems. In addition, as a result of performing venous detection of patients without body hair, 5.04% higher performance than the existing system was detected, and the proposed study results were verified. It is expected that the use of devices to which the proposed study is applied will provide more accurate vascular maps in general situations.

## 1. Introduction

Existing traditional injection methods rely on the experience and sense of the injection operator; the probability of failure increases when it is difficult to find the vein, which depends on the physical characteristics of the patient. This problem leads to a higher failure rate in patients characterized by factors that hinder the observation of blood vessels, such as infants or toddlers with small veins, Mongoloid or Negroid patients with darker skin, and obese patients with a considerable amount of fat under the epidermis.

Failure of the procedure leads to an increase in the use of disposables and economic loss at the hospital. Medical accidents can arise because of such failures, resulting in human casualties and adverse effects, and there is a pressing need for a more precise angiography technique. Techniques for ensuring the success of injection procedures that apply an image processing algorithm to the vein image with a near-infrared (NIR) camera and use a vein projector for projecting it back onto the skin surface improve vein visibility, and related research is actively underway [1,2].

Existing vein projectors apply an image processing technique to the NIR image to highlight dark regions and increase vein visibility. The vein path may be distorted by any hair noise present on the skin surface, or the noise component may be accentuated, which makes it difficult to distinguish it from the vein, although this approach can successfully accentuate vein components.

To improve this aspect, we intend to improve accuracy by removing the body hair, which is a noise element above the skin that has the greatest effect on the vein projector, from the images. In the proposed technique, a digital hair removal algorithm used to check the condition and health of existing skin is constructed and applied to remove body hair from the images [3,4].

A vein visibility enhancement algorithm is proposed in this paper. This algorithm removes hair and other noise components and enhances contrast and clarity by defining camera noise present in the vein image captured with an NIR camera and hair on the epidermis as noise. Moreover, a vein projector system is proposed to drive the proposed algorithm.

The contributions of this study can be summarized as follows:Improved results by adding a function to remove body hair that affects the image processing results.Reduction of construction cost by implementing an embedded system-based vein image processing system.Presenting a new research direction by combining digital hair removal, which has been used to check skin condition, with the field of intravenous projection.

The remainder of this paper is organized as follows. Section 2 describes related research trends on venous imaging and digital hair removal. HW specifications and structures are built in Section 3. The composition of venous visualization and digital hair removal algorithms is presented in Section 4, and performance verification results of the proposed algorithms are presented in Section 5. Finally, Section 6 describes the analysis results of the study.

## 2. Related Work

### 2.1. Vein Imaging

Vein recognition to distinguish veins in the human body relies on the practitioner’s experience in terms of discovering veins with the naked eye. Such a procedure can increase the failure rate, which can result in psychological pressure and physical pain without being provided to the patients; however, the structure of the vein can be identified, and its location can be identified through X-ray imaging by injecting a contrast medium into a vein. The time and physical cost associated with the intravenous angiography procedure are high. A vein image processing system for instantly determining the location of a vein was developed to address this issue [1,2,5]. Figure 1 shows NIR rays with wavelengths of 700–1000 nm characterized by a lower absorption rate for water, melanin, and hemoglobin.

A total of 5–10 mm veins can be observed when considering an image with an NIR wavelength that corresponds to the optical window. Skin tissue and subcutaneous fat without veins appear brighter because NIR rays that pass through veins under the skin appear darker as they absorb NIR rays. Vein image processing technology improves the visibility of images by enhancing the contrast of dark vein components and strengthening outlines using an image processing algorithm with images considered at wavelengths in the range of the optical window [2,6].

### 2.2. Related Research Trends

According to previous studies, methods to apply the digital hair removal algorithm to guarantee visibility when performing tests, such as for skin cancer and skin lesions, have been proposed [3,4,7]. Digital hair removal has been studied via various approaches, such as using simple image processing [7] and using convolutional neural networks based on deep learning [3,4].

However, existing studies focused on digital hair removal applied in general digital camera environments; studies based on NIR camera images, which are used in vein projector environments, by using a method applied to RGB images have not been reported. Therefore, we propose a digital hair removal and vein image processing algorithm that can be operated in an NIR environment to provide higher visibility when the vein is visualized.

In addition, the vein image processing algorithm used in the proposed study constructed the entire process by referring to the traditional vein image processing techniques [1,2,8] previously studied.

### 2.3. Used Algorithm

This study gathered necessary functions by analyzing and selecting existing image processing algorithms, such as morphology, Telea’s inpainting, histogram normalization, and histogram equalization, to construct the proposed algorithm. A process that enhances the vein component and removes noise hindering the visibility of the vein component was developed based on the proposed algorithm as the final result.

#### 2.3.1. Morphology

A morphology algorithm was applied to the proposed algorithm to (1) reinforce vein components and remove noise components besides vein components in the dermal layer and (2) remove hair and skin surface noises. Grayscale and binary morphology were selectively used to implement the above function. Morphological operations of binary dilation and erosion are expressed as follows:(1)$$BinaryDilation:I⊕k=\underset{x\in f}{\cup }S_{x}$$
(2)BinaryErosion:I⊖k={x|x+s∈I,∀s∈S}

In Equations (1) and (2), *I* denotes a pixel set having a value of 1, and *k* denotes a mask kernel for morphology construction. The algorithm is configured to perform dilation or erosion operation by applying the mask kernel *k* where *I* is 0 in the case of dilation and where *I* is 1 in the case of erosion. A binary morphology operation was performed to perform morphology on binary images. A closing operation was used to perform expansion and then apply development and decline again to strengthen the detected body hair component when removing body hair.

Since each algorithm is operated on grayscale rather than the binary image, the morphology is applied to values from 0 to 255 and has more complex calculation expressions. To improve the visibility of venous vessels operating based on grayscale images, grayscale morphology calculations were performed on the algorithm as a whole. Morphological operations of grayscale dilation and erosion are expressed as follows:(3)$$GrayscaleDilation:\left(I⊕k\right)\left(j,i\right)=max_{(y,x)\subseteq k}\left(I\left(j-y,i-x\right)+k\left(y,x\right)\right)$$
(4)$$GrayscaleErosion:\left(I⊖k\right)\left(i,j\right)=min_{(y,x)\subseteq k}\left(I\left(j+y,i+x\right)-k\left(y,x\right)\right)$$

In Equations (3) and (4), *I* is the image, *K* is the kernel mask, *y*, *x* are the kernel coordinates, and *j*, *i* is the image coordinates. In Equation (Equation 3), the size of the small size pixel decreases because the high pixel size increases the length based on the maximum value obtained through the max operation. Contrary to the above, Equation (Equation 4) increases the size of a pixel with a small size obtained through min stretching so that the size of a pixel with a large size decreases. Here, a small size pixel means a dark pixel, and a large size pixel represents a bright pixel. Binary Morphology and Grayscale Morphology have applied a 3 × 3 cross-shaped mask for structure generation. Figure 2 shows an image from which hair noise is removed by amplifying the outline extracted by the Blackhat operation.

#### 2.3.2. Telea Inpainting

The Telea inpainting algorithm proposed in 2004 has a function to restore the image damage and noise; the location of the noise to be restored is set based on a hair noise removal area obtained by the operation of Blackhat and morphology algorithms obtained beforehand [9]. This algorithm was developed based on the fast marching method. It starts restoring the image from the outline of the area to be restored, and it gradually proceeds with the restoration of the area inside. The pixel to be restored is replaced with a value calculated by the normalized weighted sum of the obtained pixels after finding a small range of pixels in the neighborhood of the pixel to be restored.

A pixel value that is most suitable for the region to be restored is calculated in a form wherein more weight values are provided because it is placed closer to the restoration point. Once a pixel is restored, it moves on to the next closest pixel using the fast-matching method to ensure fast operation speed. Figure 3 below shows an image restored by Telea inpainting.

#### 2.3.3. Histogram Normalization and Equalization

In this study, the visibility of the vein component was increased by maximizing the contrast between the vein and skin surface components using a histogram equalization algorithm. Further, histogram normalization was applied to obtain a clearer and sharply contrasted image by applying histogram equalization. The brightness value in the image was normalized between 0 and 255 to distribute the image histogram clustered with a specific brightness value evenly over the entire area; the equation for this is
(5)$$I_{N}=\left(I-Min\right)\frac{Max_{N}-Min_{N}}{Max-Min}+Min_{N}$$
where Max and Min represent the maximum and minimum values for the range of brightness in the current image, respectively. After normalizing the maximum and minimum values, the original value *I* is normalized by calculating $Max_{N}$ and $Min_{N}$, which are the maximum and minimum values to set the range of brightness, respectively. Figure 4 shows that the image becomes clearer and more distinct because the brightness is evenly distributed.

The contrast in an image captured by an NIR camera is poor because low-frequency components make up most of the screen; this makes it difficult to distinguish vein components with the naked eye. The histogram equalization algorithm can sharpen the image by equalizing the maximum brightness in the image. Histogram equalization is defined as
(6)$$\begin{array}{c} \begin{array}{c} h\left[i\right]=\sum_{j=0}^{i}hist\left[j\right]\\ n\left[i\right]=h\left[i\right]\times \frac{1}{n}\times I_{max} \end{array} \end{array}$$

This histogram equalization technique is used to determine the maximum brightness value based on the cumulative frequency of the brightness of pixels in each picture individually. The brightness value of each pixel in the image is replaced with the n[i] value obtained through histogram equalization to maximize the contrast.

## 3. Proposed Intravenous Projector System

The vein image processing algorithm is applied to remove the noise that distorted the path and size of the vein and improve the visibility of the vein after capturing the vein image with an NIR image sensor. The image with the improved visibility of veins is displayed over the area photographed by the projector module, which allows medical practitioners to noninvasively identify the location of veins. The entire system is configured as shown in Figure 5. At this point, based on the hardware of M’s VeinVu-100 product, the entire study was conducted by loading software and algorithms.

The embedded system is built with a small PCB board that is lightweight and miniaturized, which allows medical practitioners to adjust the position of the vein projector conveniently based on the scenario; it was developed in the C language. An embedded miniaturized intravenous projector system was used flexibly based on various medical scenarios and purposes, which can achieve several spatial advantages. The proposed embedded vein projector system proceeds with the scenario shown in Figure 6; an image processing algorithm is applied to the original image, which is then transmitted to the projector.

### 3.1. Video Shooting and Image Transfer

To secure a clean original image, the light wavelength of the near-infrared image sensor was empirically searched to find the value that best captures the vein under the epidermis. For light wavelength search, wavelengths of 700∼1000 nm were searched at intervals by adjusting each wavelength of the NIR camera; in this case, where the visibility of blood vessels is most apparent, the wavelength was adopted in the used camera band of 850 nm. The light wavelength measured by the camera employed in this paper is shown in Figure 7.

### 3.2. Video Shooting and Image Transfer

After that, calculations are performed on the PCB board for noise removal and image processing on the captured image. To optimize the load of the PCB board in charge of arithmetic processing, each frame of the image is extracted and utilized as a separate image, and each image is converted into 16-bit unsigned integer format and used.

### 3.3. Vein Image Processing and Projector Image Delivery and Transmission

Image processing is applied to the original image to enable users to check the vein with the naked eye after the vein image captured with the NIR image sensor is transferred to the PCB. The OpenCV library is used in the proposed system to ensure fast operation speed and optimized storage space. The OpenCV library is an open-source programming library developed for real-time computer vision that provides a wide variety of functions obtained through constant updates and upgrades from the past [10].

When sending and receiving images through the OpenCV library in the proposed system, the image is scaled down without distortion; this image is then improved using the image codec and processing functions. The application of vein image processing allows the image to be transmitted to the projector connected to the PCB. Then, the angle of the image projection set for the projector is adjusted to project the image on an appropriate position using a prism, and the image is transmitted to the region imaged by the NIR camera with the projector module. Finally, the resultant image of the vein location specified through image processing is projected and delivered to the user with a vein projector.

### 3.4. Building Embedded System Environment

The proposed system is built as an embedded system to lighten and optimize the vein projector to provide convenience to users by configuring the system to project veins quickly and conveniently in a medical environment. This approach enables the miniaturization of the system hardware, which helps reduce its weight. These advantages make the process of capturing a vein image, applying the vein image processing algorithm, and projecting an image with improved visibility of veins without interfering with the treatment in a medical environment smooth in addition to taking up less space.

However, the operation speed can decrease when applying the image processing algorithm because of the limited computing resources and specifications of the reduced weight and reduced hardware; this interferes with the smooth projection of the vein. Therefore, research focusing on the computation speed and optimization of each function of the vein image processing algorithm when constructing the proposed system allows us to ensure a fast computation time even with limited computing resources and to improve the quality of service.

#### 3.4.1. Cross-Compiling for System Optimization and Saving Computing Resources

The proposed system is built and operated in an embedded environment; this requires building a system to run the image processing function in a limited computing environment smoothly [11]. Image processing algorithms require more computing resources than algorithms processing data such as plain text or images. Therefore, cross-compilation, which involves compiling the source code in advance and saving it as an executable source, is performed to utilize the proposed system more quickly and comfortably by saving available resources when starting, running, and saving the system. The arm-Linux-gnueabihf compiler is used for cross-compilation; it is designed to generate the VeinProjector.o file after compilation.

#### 3.4.2. Utilization of Image Processing Library for System Optimization and Saving of Computing Resources

The system was coded based on the C++ language to utilize the limited resources efficiently; the OpenCV library was used to configure the final system to optimize the operation speed and resource consumption. The OpenCV library was developed by Intel and is actively used in the real-time computer vision field [12]. When an algorithm used for image processing is applied to an image, it is processed in one dimension rather than as a multi-array; furthermore, the speed is optimized and implemented in its own library by optimizing and applying the masking operation of the kernel. It is possible to write concise code by functionalizing various options and functions, which is advantageous in terms of compilation speed (it is significantly faster than self-made code) [10].

Only libraries including necessary functions from among all OpenCV libraries are selectively applied to cross-compilation to save computing resources and optimize the proposed system. Table 1 summarizes the OpenCV library and cross-compilation environment used in the proposed system.

## 4. Proposed Vein Image Processing Algorithm

The proposed vein image processing algorithm removes hair noise from the original image captured with an NIR camera and strengthens the vein components. To this end, it employs hair removal and vein visibility improvement algorithms. This study aimed to extract a vein image with improved blood vessel visibility by sequentially applying the specified functions to each section.

The proposed algorithm is implemented to allow the user to change between the hair removal and normal modes alternately based on the presence or absence of hair on the captured skin. When imaging an area without hair, the computation speed can be improved through efficient resource management by optimizing the computing resources wasted in the hair removal operation. Figure 8 below shows the overall flow of the vein image processing algorithm.

### 4.1. Light Component Removal

Gradation occurs in the image because of nonuniform light in the NIR image; this causes a problem in that specific areas appear darker than others [13]. The image processing algorithm may not work properly if such a problem occurs because brightness appears differently based on the angle or intensity of the light, even for pixels of the same color. Figure 9A,B show images obtained by image processing before and after removing light, respectively.

The nonuniform light in the image is removed to prevent the problem of distortion in the brightness values of objects attributed to irregular lighting to ensure the normal operation of the image processing algorithm. The reverse light component in the image is obtained to remove light from the image. The reverse component is extracted by applying the average filter with a mask size 30 × 30 or larger, which helps extract the light component of the image and calculate the difference between the maximum brightness value of the image and extracted light component. Reverse light means obtaining the inverse of the light by subtracting the image to which an average filter with a 30 × 30 kernel window is applied from the maximum value of the original image. The detailed process for this is shown in Figure 10.

### 4.2. Hair Noise Removal

Hair noise in vein image results strengthens the contrast of the hair component with the vein component when the image processing algorithm is applied; this makes it difficult to distinguish it from veins or thin blood vessels. The route of the vein may be affected, which interferes with specifying the location of the vein if the hair noise overlaps with the vein component.

The proposed system was designed to provide an option to remove hair noise for the user based on the scenario as it can make it difficult to locate the veins with the naked eye. In an image taken with an NIR camera, the hair component is photographed directly on the skin and has an outline that is sufficiently sharp to be extracted with an edge detection algorithm. The position of the hair outline on the epidermis is detected before the image processing algorithm is applied to the NIR camera image considering the characteristics of the hair outline. Then, the body hair component was changed to a binary image using a threshold value set through Otsu’s method to body hair detected through the Blackhat operation. Afterward, morphology erosion was applied to increase the scale of body hair components to obtain positional information for hair removal.

Telea inpainting is then employed to fill the hole remaining in the image with the surrounding pixel values. Figure 11 below shows the algorithm scenario applied for hair removal.

### 4.3. Vein Image Processing

A vein image processing algorithm was constructed using a clean vein image obtained through the hair noise removal algorithm. The vein image processing algorithm enhances visibility by enhancing the vein components in the image after hair noise removal.

The vein image processing algorithm constructed in this study is shown in Figure 12; the overall process is divided into six steps:Extracting and removing light components to apply image processing algorithms uniformly to the entire image;Histogram normalization and equalization for the contrast enhancement of venous components;Threshold processing to remove noise such as skin wrinkles and fat layers in addition to vein components in the image;Morphological dilation to connect disconnected vein components and remove noise;Sharpening to restore thickened vein components attributed to morphological dilation;Median smoothing for image quality improvement and small noise removal.

The brightness varies for each area in the image depending on the location of the light source when capturing a vein image; this results in an error. Furthermore, this phenomenon worsens when the image processing algorithm is applied because the contrast of the vein image is low. The reverse component to the lighting light component of the image is extracted for light removal, and thus, a vein image with uniform brightness and adequate contrast is obtained via preprocessing using histogram normalization and equalization.

In general, venous blood vessels are located below the fat layer. Although NIR penetrates fat and shoots veins, some wavelengths are lost due to imbalanced and thick grease in the penetration process, and the visibility of veins is impaired, which is similar to various noises (wrinkles, scars, etc.) In general, it has a low contrast compared to blood vessels. Therefore, to remove such noise, a threshold with a defined threshold value was applied by Otsu’s method. When the point is used, some noise with higher contrast than blood vessels in the image can be removed.

Components such as fine wrinkles and fine hairs on the skin amplified by histogram equalization and the fat layer under the skin are removed by the threshold operation. The original value was output when the value was darker than the threshold value; when it was brighter than the original value, the brightness value was output as 255 to remove noise. At this time, the threshold value was set based on the Otsu algorithm [14] to specify it according to the characteristics of the image.

Disconnected vein components were connected by morphological dilation; vein components thickened because of morphological dilation were restored to their original thickness by sharpening. In addition, the image quality was improved by reducing the size of small noise that is not a vein component. Then, median smoothing was used to obtain an image with improved quality by applying the image processing algorithm implemented above. Figure 13 below shows the images obtained through the proposed vein imaging procedure.

## 5. Intravenous Projector System Performance Evaluation

### 5.1. Experimental Evaluation Environment

The proposed system is designed based on an intravenous projector used in various environments to reduce the size and weight of each module for achieving miniaturization and lightweight characteristics while ensuring a smooth processing speed and frame. In this experimental evaluation, the performance is analyzed by measuring the processing performance of the normal and hair noise removal modes for verifying whether high-quality results are provided.

Table 2 summarizes the specifications of the PCB built into the vein projector used in the test conducted in this study.

### 5.2. Dataset

#### Dataset for Verification of Vein Image Processing Algorithm

To evaluate the performance of the proposed algorithm, a total of three experimenters’ wrists, backs of hands, and arms were photographed with NIR cameras to obtain a dataset of 20 images (10 images with body hair and 10 images without body hair).

This test added a standardized blood vessel layer for an objective comparative analysis by directly extracting and approximating the original image, including the previously photographed venous blood vessels. The blood vessel layer added to the original idea was drawn in gray with a brightness of about 57 to 62 based on grayscale to reduce the gap with the existing blood vessels. In addition, the thickness of the blood vessels was set so that the actual scale was about 0.5 to 1 cm (approximately 15 to 20 pixels based on the image); a test image was created. With SSIM [16], we performed comparative evaluation with the existing and proposed algorithms.

The proposed verification method arbitrarily analyzes blood vessels from the photographed hand with human eyes and extracts blood vessels. During comparison, objective reliability is not guaranteed and subjective judgment is injected by human observation. The blood vessel data were compared to ensure objectivity. The detailed algorithm verification method on the generated images is shown in Figure 14.

### 5.3. Functional Evaluation of Suggested Algorithm

#### 5.3.1. Proposed Algorithm Accuracy Verification

The proposed system provided a hair removal function for improving the visibility of the image processing results for vein images containing a considerable amount of body hair. In this test, the performance was measured to verify the effectiveness by evaluating the degree to which hair removal improved the original image containing body hair.

In addition, the original image and dataset for verifying the hair removal mode in which hair noise is added to the original image were applied to the hair removal algorithm, respectively, and the output results are compared. The resultant image obtained by inputting the original image was compared with the resultant test image to which body hair was added, and it was checked whether distortion and deformation of the vein component due to the removal of body hair occurred.

To evaluate the performance of the proposed algorithm, comparison verification was performed with the algorithm installed in the ‘Veinvu 100’ product developed by Company M. The performance verification of the algorithm was conducted by comparing the output image and blood vessel image inserted in Figure 14 through SSIM to measure the similarity. The detailed verification method is as follows:To clearly distinguish the location of the vein output from each algorithm, it is converted into a binary image by applying a threshold to the resulting image.The similarity between binary images and expert-marked vein location images is compared using SSIM.The average of the similarities obtained through comparison of images of the entire data set is calculated.

#### 5.3.2. Performance Verification of Vein Image Processing Algorithm Based on Hair Noise Removal

This test was conducted to determine the degree to which the proposed body hair noise removal algorithm improves the performance of the vein image processing algorithm. Based on the original image including body hair noise, Figure 15D was obtained, and the performance of the proposed algorithm was verified with a total of 10 images.

As a result of the test, it was possible to extract a clearer and more distinct vein path than with the vein image processing algorithm installed in Veinvu-100, and the results are shown in Figure 16. Figure 16 was constructed by randomly selecting three images from among the images used in this verification.

As can be seen in Figure 16, in (C) overall, compared to (D), the effect of body hair noise was reduced, the contrast of blood vessels was clearer, and the cutoff or omission of blood vessels was reduced. This can be seen from the histogram of (C). In the case of (C), components other than blood vessels are removed to maximize contrast with blood vessels. A histogram distribution similar to image (B), including only the blood vessel layer, can be confirmed. On the other hand, in (D), the contrast of body hair is the strongest. As a result, the visibility of blood vessels with relatively weak contrast is somewhat less improved. It can be seen that the histogram distribution is also similar to the original image.

For objective performance verification between the proposed and Veinvu-100 algorithms, SSIM-based similarity evaluation was performed using the output images of each algorithm.

In Figure 16C, compared to (D), it can be confirmed that the overall noise was sufficiently reduced to be clearly identified with the naked eye. As a result of checking the similarity for all 10 images used in the test, the proposed algorithm measured an average SSIM of 74.93%, and the VeinVu-100 algorithm had an average of 64.55%, indicating that the proposed algorithm improved performance by 10.38% compared to VeinVu-100. Detailed verification results can be seen in Figure 17 and Table 3 below.

#### 5.3.3. Simple Vein Image Processing Algorithm Performance Verification

This test was conducted to verify the performance of the proposed vein image processing algorithm itself. Figure 18D was obtained based on the original image without body hair noise, and the performance of the proposed algorithm was verified with a total of 10 images.

As a result of the test, in Figure 19C, output by the proposed algorithm, and in (D), output by the VeinVu-100 algorithm confirm the extent to which the path and thickness of the vein can be clearly identified. However, in the case of (C), the brightness of non-venous locations was significantly different from that of veins, and overall, noise was reduced. This makes it easy to grasp the length and thickness of the vein at a glance.

In the case of the original image (A), it can be seen that the histogram is relatively crowded due to the imbalance of lighting and weak contrast between each component. However, in the proposed algorithm (C), only vascular details with apparent differences were obtained by enhancing the visibility of blood vessels and removing noise components. Therefore, when confirmed with a histogram, it has a form similar to (B). In the case of (D), the contrast with the blood vessels was improved by weakening the vital noise component through image processing, and the blood vessels were captured generally because there was no body hair. However, the histogram has various contrast ratios due to noise detected from components other than blood vessels.

For objective performance verification between the proposed and Veinvu-100 algorithms, SSIM-based similarity evaluation was performed using the output images of each algorithm.

As a result of examining the similarity for all 10 images used in the test, the average similarity of the proposed algorithm was 86.52%, and that of the VeinVu-100 algorithm was 81.48%, indicating that the proposed algorithm improved performance by 5.04% compared to VeinVu-100. Detailed verification results can be seen in Figure 20 and Table 4 below.

#### 5.3.4. Vein Image Processing Algorithm Speed Performance Verification

The calculation time was measured to check whether the proposed system built in the embedded environment can provide a smooth frame image. To verify this, the number of pixels corresponding to each body hair region was counted for 30 different test images and compared with the measured calculation time. As a result of checking the measured computational speed, the more body hair in the image, the slower the computational speed due to the characteristic of the inpainting algorithm, which increases the computational amount in proportion to the proportion of body hair. For this reason, in the case of the vein image processing algorithm operating without the hair removal function, an average of 24 ms was measured, whereas when the hair removal function was included, the calculation time increased by about 88 ms as the proportion of body hair increased by about 10%. The results of this can be found in Figure 21.

## 6. Conclusions and Discussion

### 6.1. Conclusions

A vein projector system that could cope with more diverse noise environments than previously proposed vein image processing algorithms was proposed in this paper. The proposed system utilizes the OpenCV library and uses the image processing algorithm API to transmit images smoothly. It converts the C++ source code into an executable file through cross-compilation to enable fast operation in systems with limited computing resources.

In addition, from a performance perspective, when the hair removal mode is used, body hair noise that adversely affects the performance and accuracy of the vein image processing algorithm, such as distorting the vein path or reducing visibility, is removed without distortion of the vein component. Therefore, when there is significant body hair noise in the image, the adverse effect on the veins can be improved and vein distortion can be excluded.

To objectively verify the performance of the proposed algorithm, a comparative evaluation was performed with M company’s VeinVu-100 algorithm, which is an existing product. As a result of SSIM-based similarity measurement, when the performance of the proposed algorithm was confirmed in the environment of photographing skin with significant body hair if body hair is not removed, it causes image distortion during the entire calculation process, which is useful for distinguishing between body hair noise and vein components. Meanwhile, when the proposed algorithm was used, it was confirmed that the visibility of the vein was improved by minimizing distortion or cracking, and the degree of improvement was sufficient to be clearly confirmed with the naked eye.

As a result of comparison in an environment without body hair noise, the similarity of the proposed algorithm was measured on average to be 5.04% higher than that of the comparison algorithm. In addition, as a result of measuring the similarity in an environment including body hair noise, it was 10.38% higher than that of the comparison algorithm, confirming that the performance was improved by approximately 5.34% through the hair noise removal algorithm.

### 6.2. Discussion

The proposed system can be applied to real healthcare settings, where it can provide adaptive responses to various noise environments unlike existing vein image processing algorithms that suffer from operation difficulties under various noise conditions that obstruct the visibility of the vein and distort the vein path; furthermore, it can also perform the conventional role of an existing vein image processing algorithm. In addition, it is expected that the method and approach based on image detection, the core subject of this study, can be utilized for various medical image detection and classification [17,18].

However, for this, it is necessary to further guarantee the accuracy and safety of the proposed algorithm by conducting clinical tests to verify whether it is practically applicable to the field. Thus, clinical tests will be conducted in collaboration with hospitals and experts.

In addition, when the inpainting algorithm is applied in the process of removing body hair, if there is tissue such as fat, scar, or muscle inside the skin at the boundary line, there is a problem in that it acts as a noise component. It is judged that this problem requires additional research to improve the inpainting algorithm to minimize the noise generated by it.

## Figures and Tables

**Figure 1 sensors-22-08559-f001:**
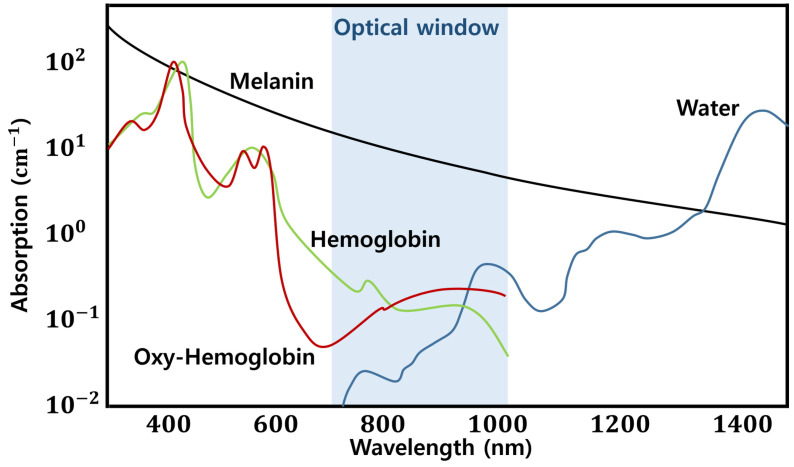
Graph for the absorption rate of water, melanin, and hemoglobin according to the wavelength of light.

**Figure 2 sensors-22-08559-f002:**
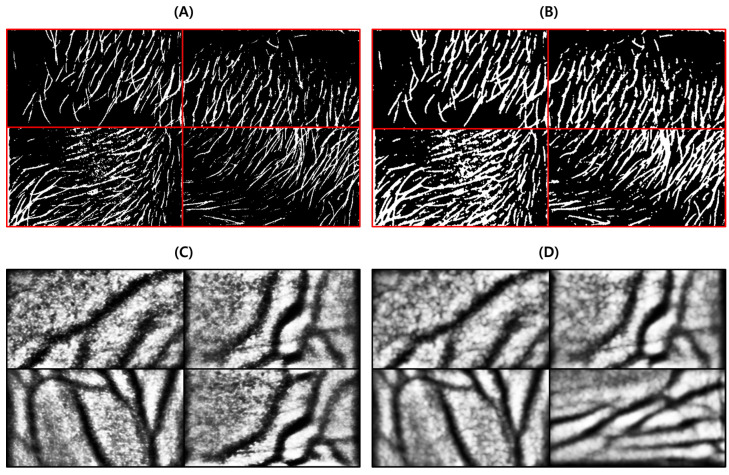
Results of applying morphological operations to the outline: (**A**) is the result of extracting body hair position (Blackhat), (**B**) is the result of removing noise and amplifying body hair by applying binary morphology dilate and open, and (**C**) is an image after histogram smoothing, (**D**) is an image obtained by applying an average filter and grayscale morphology operation to remove noise and perform operations for vein enhancement.

**Figure 3 sensors-22-08559-f003:**
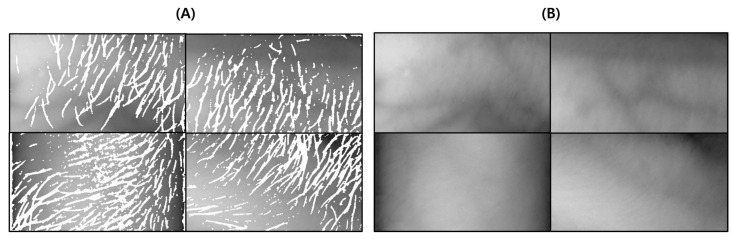
Vein image restored by Telea inpainting: (**A**) is an image showing body hair, and (**B**) is an image with Telea inpainting applied.

**Figure 4 sensors-22-08559-f004:**
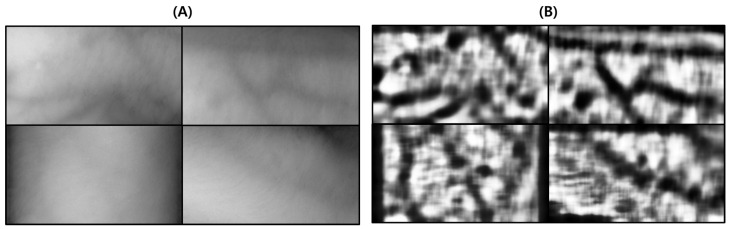
Vein images with histogram normalization and equalization applied: (**A**) is the original image, and (**B**) is the result of applying histogram equalization. The contrast of venous vessels became clear through histogram smoothing.

**Figure 5 sensors-22-08559-f005:**
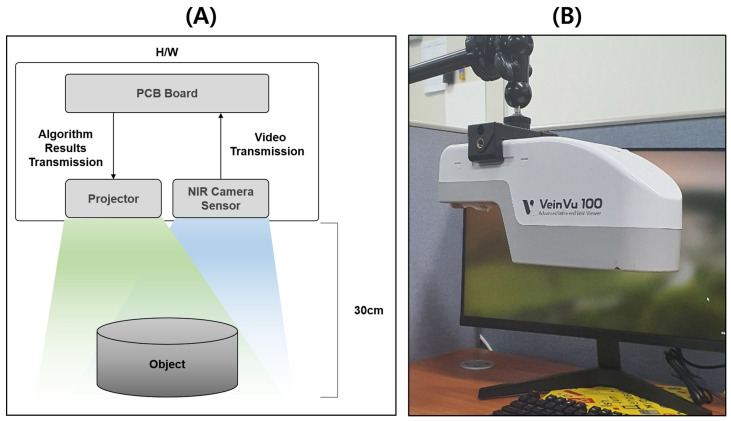
Structure of embedded vein projector: (**A**) is the structure of VeinVu-100 used in the proposed study, and (**B**) is an image of the device used.

**Figure 6 sensors-22-08559-f006:**
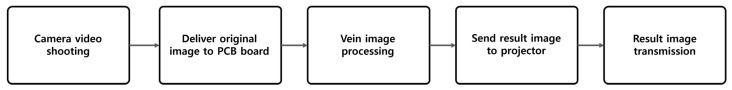
Proposed vein image processing system scenario.

**Figure 7 sensors-22-08559-f007:**
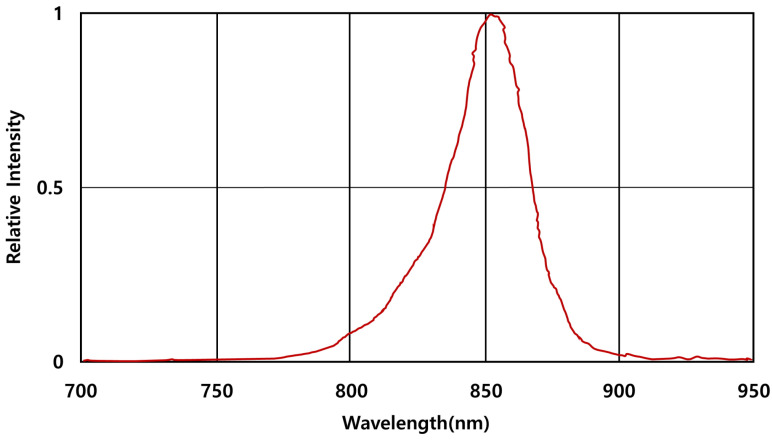
Graph of light wavelength measured by NIR camera.

**Figure 8 sensors-22-08559-f008:**
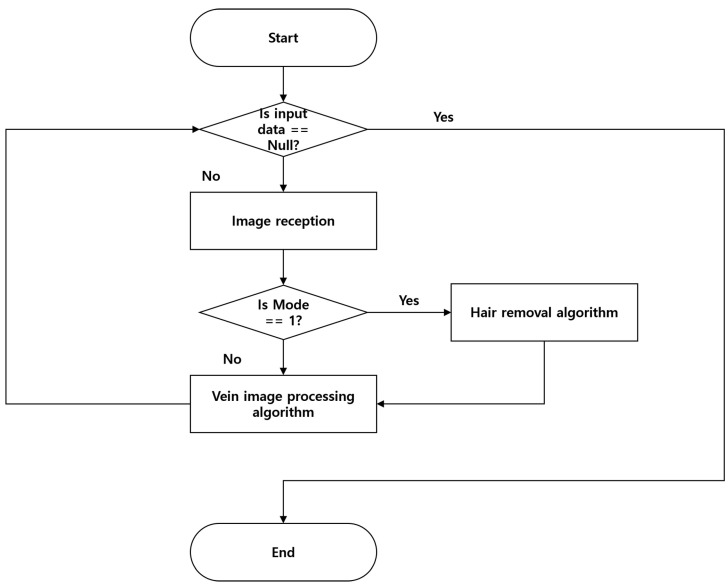
Flowchart of the proposed vein image processing algorithm.

**Figure 9 sensors-22-08559-f009:**
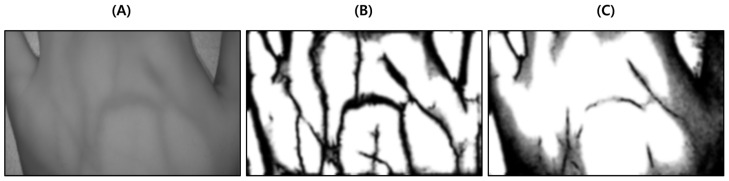
Comparison of image processing results with and without removing light: (**A**) is the original image, (**B**) has unbalanced lighting removed so the applied operations can usually work and strengthen the veins. (**C**) shows that the brightness of the unbalanced illumination is relatively darker than that of the vein, so the calculations are applied to the darkly illuminated area. This eliminates the contrast of veins.

**Figure 10 sensors-22-08559-f010:**
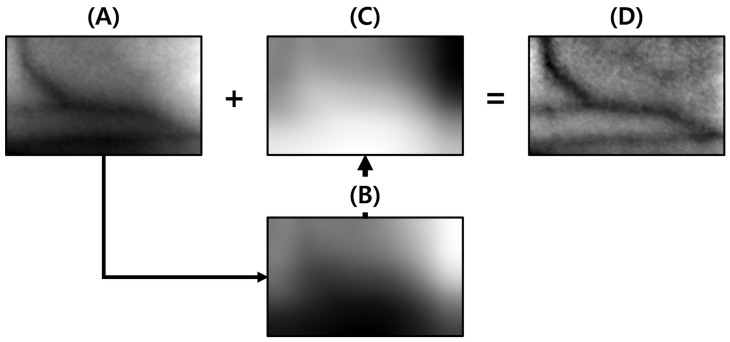
(**A**) is the original image, (**B**) is the background approximation by blurring, (**C**) is the inverted image of (**B**), and (**D**) is the background approximation based on summing (**A**,**C**) resulting an image with the lighting removed.

**Figure 11 sensors-22-08559-f011:**
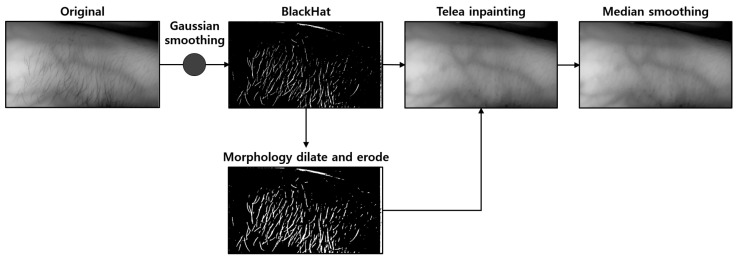
Hair removal algorithm scenario.

**Figure 12 sensors-22-08559-f012:**
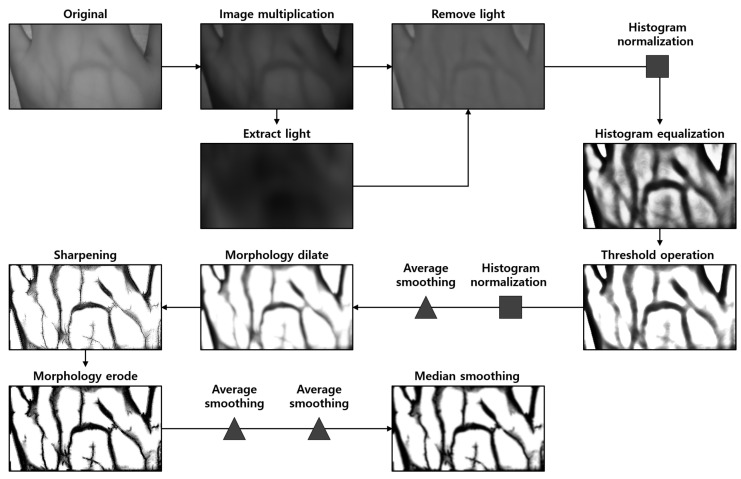
Vein image processing algorithm scenario.

**Figure 13 sensors-22-08559-f013:**
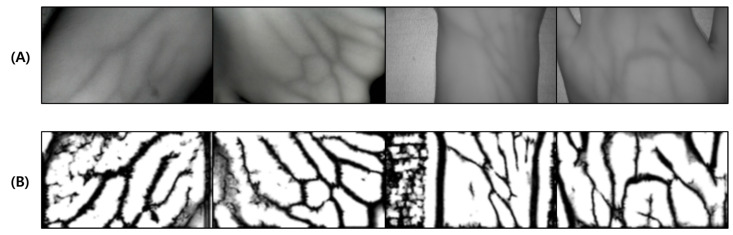
Images obtained by applying the vein image processing algorithm: (**A**) is the original image taken with an NIR camera, and (**B**) is the resultant image obtained through the proposed algorithm.

**Figure 14 sensors-22-08559-f014:**
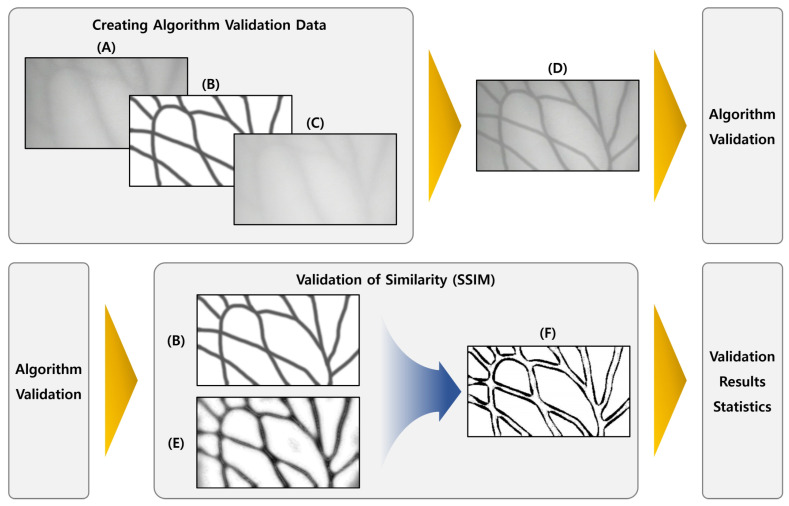
Algorithm verification method using the dataset for verification of the vein image processing algorithm: (**A**) is the original image, and (**C**) is the original image with 85% transparency applied. (**B**) is an artificial blood vessel layer added for verification, and (**D**) is a composite image of (**A**–**C**). (**E**) is the result image obtained through the algorithm, and (**F**) is the result image obtained by comparing (**B**,**E**) with SSIM.

**Figure 15 sensors-22-08559-f015:**
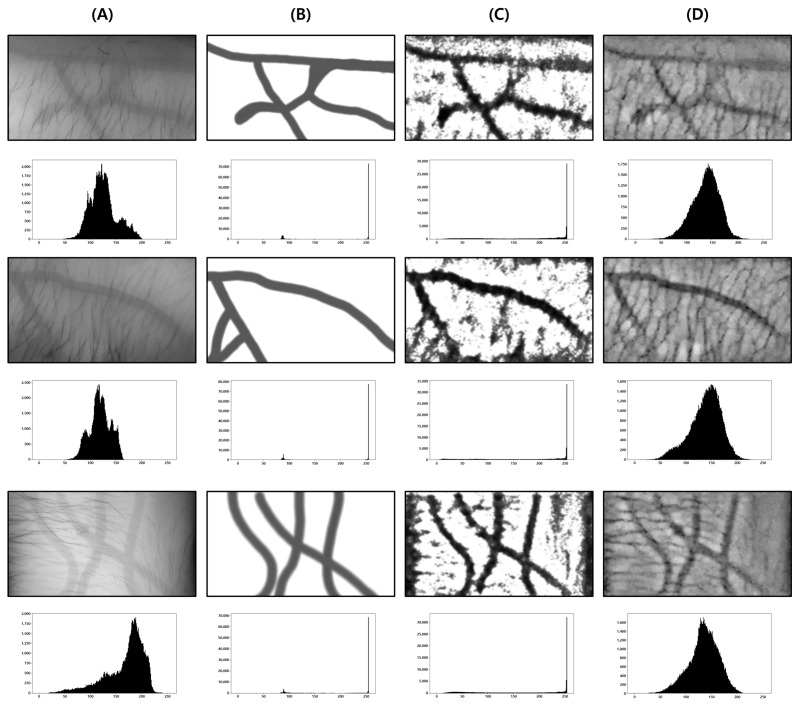
Image operation results with body hair noise: (**A**) is a test image created for algorithm verification, and (**B**) is an artificial blood vessel layer used to create the image. (**C**) is the resultant image of the proposed algorithm including the hair noise removal function, and (**D**) is the resultant image of the Veinvu-100 algorithm.

**Figure 16 sensors-22-08559-f016:**
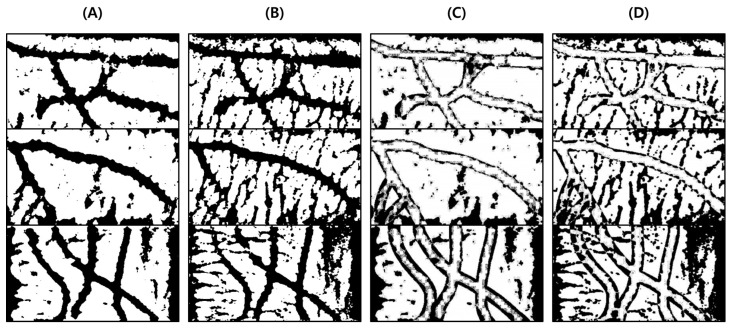
Result of applying SSIM to the results of the proposed algorithm and Veinvu-100 algorithm: (**A**,**B**) are binary images obtained by applying the proposed and Veinvu-100 algorithms, respectively, and (**C**,**D**) are the results of comparing (**A**,**B**) with the vascular layer using SSIM.

**Figure 17 sensors-22-08559-f017:**
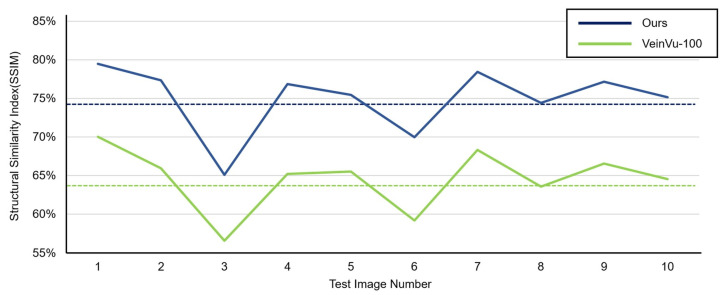
Proposed Algorithm Operation SSIM Measurement Results.

**Figure 18 sensors-22-08559-f018:**
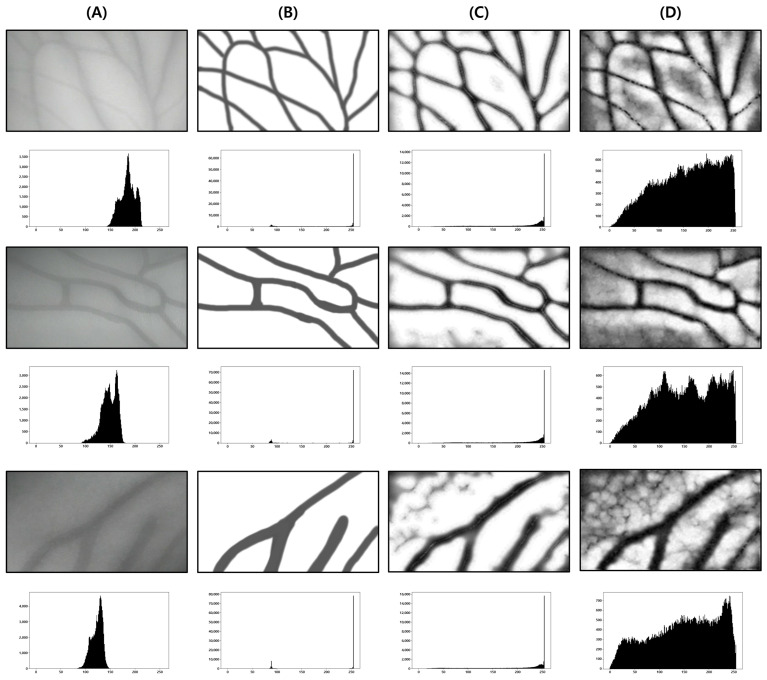
Image operation results without body hair noise: (**A**) is a test image created for algorithm verification, and (**B**) is an artificial blood vessel layer used to create the image. (**C**) is the resultant image of the proposed vein image processing algorithm, and (**D**) is the resultant image of the Veinvu-100 algorithm.

**Figure 19 sensors-22-08559-f019:**
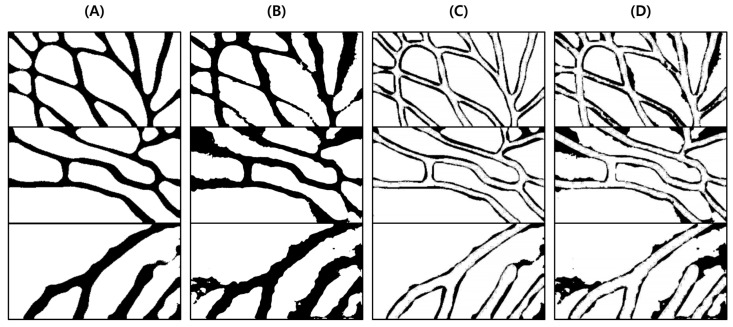
Results of applying SSIM to the results of the proposed and Veinvu-100 algorithms: (**A**,**B**) are binary images of the results of applying the proposed and Veinvu-100 algorithms, respectively, and (**C**,**D**) are results of comparing (**A**,**B**) with the vessel layer using SSIM.

**Figure 20 sensors-22-08559-f020:**
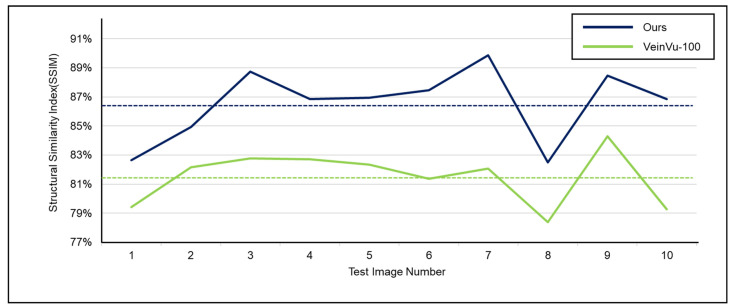
Proposed Algorithm Operation SSIM Measurement Results.

**Figure 21 sensors-22-08559-f021:**
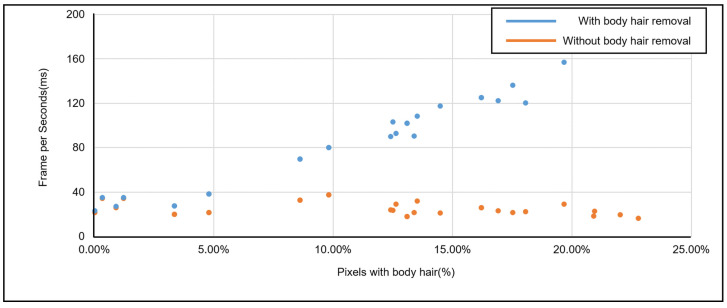
Proposed Algorithm Operation Speed Measurement Results.

**Table 1 sensors-22-08559-t001:** Cross-compilation environment for system optimization.

Item	Specifications
Compiler	arm-Linux-gnueabihf-g++
OpenCV version	OpenCV 3.4.1
Included library	opencv2/opencv
	opencv2/imgproc
	opencv2/video
	opencv2/core
	opencv2/imgproc
	opencv2/highgui

**Table 2 sensors-22-08559-t002:** PCB performance table for testing [15].

Item	Specifications
CPU	Quad core ARM Cortex-A53 CPU Broadcom (1.4 GHz)
RAM	1GB LPDDR2 SDRAM
GPU	Video core IV
Storage	128 GB SD Card
OS	Raspbian STRETCH LITE (1.8 G)
LAN	2.4 GHz and 5 GHz IEEE 802.11.b/g/n/ac Wireless LAN and Bluetooth 4.2/BLE
Size	120 mm × 75 mm × 34 mm
Weight	75 g

**Table 3 sensors-22-08559-t003:** Proposed Algorithm SSIM Similarity Verification Results.

Item		Ours	Veinvu-100
Images with hair	Max	79.46%	70.03%
	Min	65.07%	56.59%
	Average	74.93%	64.55%

**Table 4 sensors-22-08559-t004:** Proposed Algorithm SSIM Similarity Verification Results.

Item		Ours	Veinvu-100
Images without hair	Max	89.86%	84.30%
	Min	82.49%	78.38%
	Average	86.52%	81.48%

## Data Availability

Not applicable.

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
