# Peer review of "Design and Implementation of Embedded-Based Vein Image Processing System with Enhanced Denoising Capabilities"

_sensors, 2022, doi:10.3390/s22218559_

Round 1
Reviewer 1 Report
- Equation (1) and (2): Please explain what the variables are in the equation. ( I suppose I is image, k is kernel, i and j are indices, but it is not clear if you do not explain about it). - The symbol you used for erosion is not correct. You have used same symbol for both dilation and erosion - In Figure 2, I assume you applied erosion followed by dilation as morphological image opening. But you haven’t explained it. Dilation followed by erosion is also possible. So it should be clarified. - In Figure 2, I assume you applied the morphological image operations on the binary image not gray scale image. Please clarify this too. If you used binary image please clarify the way you selected the threshold for converting the image to binary image. - Your definition of dilation and erosion are not clear. Please cite any reference you used for definition. Especially when you mentioned: “make large values larger and small values smaller” and also “make large values smaller and small values larger” - In lines 158 and 159, I think you need to replace PBC by PCB Section 4.1 Light component removal: - What is reverse light component? - What do you mean applying mask to the average filter? (I guess you mean computing average value over a sliding window of size 30 x 30). But the explanation is not clear! - The sentence “The light component present in the image is removed by incorporating the extracted reverse component with the original image.” is vague! What exactly do you mean by “incorporating”?
Author Response
검토해 주셔서 감사합니다. 고객님의 댓글과 조언을 꼼꼼히 확인하여 글을 수정하였습니다. 자세한 사항은 첨부파일에서 확인하실 수 있습니다.

Reviewer 2 Report
The author's paper "Design and Implementation of Embedded-based Vein Image Processing System with Enhanced Denoising Capabilities "describes the algorithm they proposed to enhance vein images and the improvements they made to the imaging system, and verifies the feasibility of the proposed algorithm through some practical tests. It makes a good exploration for the work of vein image assisted precision medicine. In view of the content, science and standardization of the paper, the author should consider and modify the following questions:
1. In the section "Video Shooting & Image Transfer" in Section 3.1 of the paper, whether the search for the best vein imaging wavelength based on experience can be described in more detail and the process of selecting light wavelength is briefly described;
2. In this paper, it is suggested that the difference between some images can be observed by naked eye, whether third-party reference can be added, or the contrast and other feature information of the image can be quantified to make it more intuitive and convincing;
3. In the picture annotation, it is possible to add some explanatory content. For example, can you explain the differences of several pictures in picture 8 and make a simple summary of the reasons for the differences, which is more convenient for readers?
4. Whether the title in Table 3 is not completely consistent with the content in the table, and whether it can be expressed in the form of treatment method or sample parameter selection.
5. References refer to the latest international published research results.

Author Response
Thank you for your review. I have revised the text by carefully checking your comments and advice. Details can see in the attachment.

Reviewer 3 Report
The main contribution and proposed approach have some novelty in contribution. Revision in terms of technical details is needed before publication. So, some comments are suggested to describe technical details.
1. It is suggested to discuss about the runtime of your proposed approach briefly. (Comparison with other methods is not needed)
2. Discuss about the “fat layers” with more details
3. Dilation operator can be performed with different structure elements in different sizes. Add more details about morphological operators which are used in your proposed approach.
4. Your proposed approach can be used in medical application such as DNA analysis. For example, I find a paper titled “DNA Repair Genes (APE1 and XRCC1) Polymorphisms–Cadmium Interaction in Fuel Station Workers”, which has enough relation. Cite this paper and discuss about it briefly as advantage of your proposed approach.
5. What is the difference between “Blood vessel thickness and blood vessel color code”? Discuss briefly.
6. Add related reference for “Veinvu-100 method” in the Table 5.
Author Response

(The authors gave the same response as above.)

Round 2
Reviewer 1 Report
revised paper looks good.
Reviewer 3 Report
The authors' answers to most of the questions are convincing. Comments are considered in the revised version. Descriptions have been added to the text that better explain the presented method.